Benthic communities at two remote Pacific coral reefs: effects of reef habitat, depth, and wave energy gradients on spatial patterns

Williams Gareth J. 1 gareth@ucsd.edu
Smith Jennifer E. 1
Conklin Eric J. 2
Gove Jamison M. 3 4
Sala Enric 5 6
Sandin Stuart A. 1
1 Scripps Institution of Oceanography, University of California , San Diego, La Jolla, CA , USA
2 The Nature Conservancy, Honolulu, Hawaii , USA
3 Coral Reef Ecosystem Division, Pacific Islands Fisheries Science Center , USA
4 Joint Institute for Marine and Atmospheric Research, University of Hawaii at Manoa, Honolulu, Hawaii , USA
5 National Geographic Society , Washington, DC , USA
6 Centre d’Estudis Avançats de Blanes (CSIC) , Blanes , Spain
Bruno John
Electronic publication date: 2013 May 28
Publication date: 2013
Volume: 1
Electronic Location ID: e81
Received 2013 Apr 22; Accepted 2013 May 12
Copyright: © 2013 Williams et al.
Copyright year: 2013
Copyright holder: Williams et al.
License: This is an open access article distributed under the terms of the Creative Commons Attribution License, which permits unrestricted use, distribution, and reproduction in any medium, provided the original author and source are credited.
License URL: https://creativecommons.org/licenses/by/3.0/

Keywords: Coral, Macroalgae, Zonation, Wave exposure, Kingman Reef, Palmyra Atoll, Spatial clustering, Wave impacts, Benthic competition

Funding: National Geographic Society Moore Family Foundation Fairweather Foundation Medical Foundation for the Study of the Environment Scripps Institution of Oceanography (SIO) Gordon and Betty Moore Foundation This work was funded by the National Geographic Society, the Moore Family Foundation, the Fairweather Foundation, the Medical Foundation for the Study of the Environment, Ed Scripps, John Rowe, and another private donor. GJW was partly supported by a Scripps Institution of Oceanography (SIO) post-doctoral fellowship and is now supported by a grant from the Gordon and Betty Moore Foundation. The funders had no role in study design, data collection and analysis, decision to publish, or preparation of the manuscript.

==============================
Kingman Reef and Palmyra Atoll in the central Pacific are among the most remote coral reefs on the planet. Here we describe spatial patterns in their benthic communities across reef habitats and depths, and consider these in the context of oceanographic gradients. Benthic communities at both locations were dominated by calcifying organisms (54–86% cover), namely hard corals (20–74%) and crustose coralline algae (CCA) (10–36%). While turf algae were relatively common at both locations (8–22%), larger fleshy macroalgae were virtually absent at Kingman (<1%) and rare at Palmyra (0.7–9.3%). Hard coral cover was higher, but with low diversity, in more sheltered habitats such as Palmyra’s backreef and Kingman’s patch reefs. Almost exclusive dominance by slow-growing Porites on Kingman’s patch reefs provides indirect evidence of competitive exclusion, probably late in a successional sequence. In contrast, the more exposed forereef habitats at both Kingman and Palmyra had higher coral diversity and were characterized by fast-growing corals (e.g., Acropora and Pocillopora), indicative of more dynamic environments. In general at both locations, soft coral cover increased with depth, likely reflecting increasingly efficient heterotrophic abilities. CCA and fleshy macroalgae cover decreased with depth, likely due to reduced light. Cover of other calcified macroalgae, predominantly Halimeda, increased with depth. This likely reflects the ability of many calcifying macroalgae to efficiently harvest light at deeper depths, in combination with an increased nutrient supply from upwelling promoting growth. At Palmyra, patterns of hard coral cover with depth were inconsistent, but cover peaked at mid-depths at Kingman. On Kingman’s forereef, benthic community composition was strongly related to wave energy, with hard coral cover decreasing and becoming more spatially clustered with increased wave energy, likely as a result of physical damage leading to patches of coral in localized shelter. In contrast, the cover of turf algae at Kingman was positively related to wave energy, reflecting their ability to rapidly colonize newly available space. No significant patterns with wave energy were observed on Palmyra’s forereef, suggesting that a more detailed model is required to study biophysical coupling there. Kingman, Palmyra, and other remote oceanic reefs provide interesting case studies to explore biophysical influences on benthic ecology and dynamics.

Introduction

On many coral reefs, combinations of natural and anthropogenic forcings interact to influence benthic dynamics (Grigg, 1995). For example, nutrient pollution can fuel algal growth and influence competition between corals and algae for space (McCook, 1999), fishing removes key fish species, such as herbivores, that maintain algal standing stocks (Jackson et al., 2001), and coastal development increases sedimentation, leading to smothering and death of corals (Fabricius, 2005). While natural forcings are still present (Hughes & Connell, 1999), their relative influence on benthic community dynamics on a given reef may vary depending on the magnitude of anthropogenic disturbances (Barott et al., 2012). These interactions make it challenging to discern the independent effects of human-induced versus natural forcings on coral reef dynamics and community organization.

Throughout the Pacific Ocean there are numerous remote islands and atolls (Maragos & Williams, 2011). These remote reefs lack local human impacts and are often characterized by high fish biomass (Williams et al., 2011c) and benthic communities dominated by hard corals and other calcifying (reef-building) organisms (Sandin et al., 2008; Vroom et al., 2010; Page-Albins et al., 2012). In the absence of local human impacts, intra-island variation in reef community organization is likely a result of natural variations in predation, competition for space, gradients in physical forcings such as light and wave energy, and disturbance events, such as storms (Hughes, 1989; Rogers, 1993; Hughes & Connell, 1999). For example, the decrease in irradiance with depth is a crucial factor limiting the distribution of autotrophic organisms such as algae (Huston, 1985). The distribution of scleractinian (stony) corals is also structured by depth, but the patterns are often more complex (Done, 1983). Like algae, corals receive most of their energy from photosynthesis via their symbiotic zooxanthellae (Muscatine & Porter, 1977), though most colonies, particularly zooxanthellate soft corals, actively supplement themselves through heterotrophy (Fabricius & Dommisse, 2000). Thus, corals have more trophic flexibility than algae and are less constrained in their depth distributions. Particulate food supply often increases with depth (i.e., with increased proximity to sources of upwelling), creating a nutrient and energy-rich environment for coral growth (Leichter & Salvatore, 2006). However, high oscillatory flow in the shallows may also supply a high flux of particulate food and nutrients (Sebens & Johnson, 1991).

Other physical forcings, such as wave energy gradients, also play a major role in species zonation and benthic community organization on coral reefs (Bradbury & Young, 1981; Done, 1983; Dollar & Tribble, 1993). Extreme wave energy in shallow waters may reduce overall coral cover and favor communities dominated by more wave-tolerant growth forms with greater structural integrity (Dollar, 1982; Madin & Connolly, 2006). These communities may remain characterized by early colonizers, as repeated disturbances prevent the establishment of a late-successional climax community (Grigg, 1983; Hughes & Connell, 1999). Because algae are also vulnerable to physical dislodgement (Engelen et al., 2005), these communities may also shift to more wave-tolerant species and morphologies, such as encrusting and turf algae, with increased wave energy.

Kingman Reef and Palmyra Atoll, approximately 1300 km south of Hawaii in the northern Line Islands, central Pacific, are among the most remote coral reefs on the planet. These reefs represent a biodiversity hotspot in the central Pacific (Maragos & Williams, 2011), and previous expeditions have documented their high fish biomass and live coral cover (Sandin et al., 2008; Friedlander et al., 2010; Kenyon, Maragos & Wilkinson, 2010) and diverse algal assemblages (Tsuda, Fisher & Vroom, 2012). Kingman and Palmyra have been protected as National Wildlife Refuges by the US Fish and Wildlife Service since 2000 and 2001, respectively. In 2009, both became part of the Pacific Remote Islands Marine National Monument, further reinforcing their high-level conservation status (Kenyon, Maragos & Vroom, 2012). Here, we use Kingman and Palmyra as case studies to shed light on the natural history of benthic community patterns on remote oceanic reefs in the Pacific. Our aim was to provide a comprehensive description of benthic community patterns across reef habitats, depths and, in particular, across horizontal wave energy gradients.

We first compare and contrast benthic communities between more sheltered and environmentally stable reef habitats, to those habitats more exposed to open ocean conditions and therefore punctuated disturbance events, such as large swells. Secondly, we examine benthic community shifts in response to increases in depth; specifically, whether relationships exist between depth and: (1) the cover of macroalgae (e.g., decrease in cover with increases in depth following reduced light availability), (2) the cover of hard coral (e.g., peak at intermediate depths where there is a balance between food availability and levels of physical disturbance), and (3) the cover of soft corals (e.g., increase in cover with depth due to increasingly efficient heterotrophic abilities). Lastly, we examine how horizontal gradients in wave energy influence various aspects of benthic community organization; specifically, whether relationships exist between wave energy and: (1) hard coral cover (e.g., decrease in cover with increasing wave energy), (2) spatial clustering of corals (e.g., higher clustering with increasing wave energy due to selection for more wave-tolerant species in pockets of localized shelter), (3) algal cover (e.g., generally more or less algae present in areas of higher wave energy), and (4) algal community structure (e.g., more wave-tolerant morphologies with increasing wave energy).

Materials and methods

Study sites

Data were collected at Kingman Reef National Wildlife Refuge (NWR) (6.4 ° N, 162.4 ° W) in 2005 (Sandin et al., 2008) and 2007 and at Palmyra Atoll NWR (5.9 ° N, 162.1 ° W) in 2010. Kingman and Palmyra are approximately 60 km apart and are the two northernmost reefs in the Line Islands chain (Fig. 1). Kingman, lacking emergent vegetated dry land, is an atoll reef, whereas Palmyra supports numerous vegetated islets and is a true atoll (hereafter, both are called atolls) (Maragos & Williams, 2011). The reefs of Kingman and Palmyra cover approximately 48 km2 and 52 km2, respectively (depth range = 30 m to shoreline). Neither Kingman nor Palmyra has ever had a permanent resident human population. During WWII-era temporary occupation by the US Navy on Palmyra several alterations were made, including lagoon dredging and causeway construction that altered water flow in and out of the lagoon and shifted patterns of coastal erosion and sedimentation (Collen, Garton & Gardner, 2009). Palmyra’s lagoon is now heavily degraded, characterized by high turbidity, sedimentation, and a benthos dominated by sponges, not corals (Knapp et al., 2012) (Fig. 3H). Present-day direct human impacts at Palmyra are minimal to non-existent; only a small team of scientific researchers and staff (4–20 people) are resident at the on-island research station.

Figure 1 Map of survey sites.

Location of Kingman Reef and Palmyra Atoll, northern Line Islands (A) and the distribution of survey sites at Kingman in 2007 (B) and at Palmyra in 2010 (C). 05; indicate sites at Kingman surveyed in 2005. Numbers in parentheses refer to the depths (m) surveyed within each reef habitat.

Benthic community surveys

Study sites were chosen using a stratified random design within each of the major reef habitats found at Kingman and Palmyra (Fig. 1). Although reef flats are present on both atolls, their shallow and exposed nature made surveying them unsafe, thus no sampling occurred here. To test the effect of habitat on benthic community patterns (while keeping depth constant), surveys were conducted at 10 m depths across all three reef habitats found at Kingman (patch reef, backreef, and forereef). Reef habitats at Palmyra varied greatly in their depth ranges, confounding any explicit test of habitat. To test the effect of depth (while keeping habitat constant), surveys were conducted at 5, 10 and 20 m on the forereef at both Kingman and Palmyra. We also tested the effect of depth on the reef terrace at Palmyra by surveying at 5 and 20 m; we did not obtain adequate replication at the 10 m depth strata (n = 2) to include this information. Finally, we surveyed a unique shallow (<3 m) backreef habitat in Palmyra’s northeast (Fig. 1). Along each transect, percent cover of benthic organisms was calculated using a modification of the photoquadrat method (Preskitt, Vroom & Smith, 2004). At Kingman, two 25 m transects (separated by 10 m) were placed on the benthos at each depth at each site, while at Palmyra a single 50 m transect was used. At Kingman, 10 photoquadrats were captured adjacent to the transect line at fixed intervals, totaling 20 photoquadrats per site. At Palmyra, the number varied spatially but generally equaled 10–20 photoquadrats at fixed intervals along each transect (Table S1). Each photograph captured an area of 0.54 m2 in 2005 and 0.63 m2 in 2007 and 2010. Image analysis was completed using Photogrid 1.0. One hundred points were placed over each photograph in a stratified random design. Organisms under each point were identified to the genus level, with some categories later combined in order to standardize suspected variations in taxonomic identification (e.g., the zooxanthellate soft corals Cladiella, Lobophytum, and Sinularia were combined into “leather coral”). Turf algae (including the “epilithic algal matrix”) were defined as a mixed community of filamentous algae and cyanobacteria generally less than 2 cm tall. Crustose coralline algae (CCA; multiple genera) were identified to functional group. Calcified macroalgae were separated from fleshy macroalgae by the presence of calcium carbonate (CaCO3); all taxa that contained CaCO3 were considered calcified (these were principally species of Halimeda and Peyssonnelia). All benthic organisms that produce calcium carbonate were combined to define total calcified cover. Raw percent cover data for all habitats and depth strata are reported in Table S1.

Wave energy

We consider the effects of wave energy by comparing spatial patterns of benthic communities across horizontal gradients in wave power, calculated as: ρg264πHs2Tp

where ρ is the density of seawater (1024 kg m-3), g is the acceleration of gravity (9.8 m s-2), Hs is mean significant wave height (m), and Tp is the dominant wave period(s). To quantify wave energy (kW/m), we used NOAA’s Wave Watch III (WWIII; http://polar.ncep.noaa.gov/waves); a global, full-spectral wave model. A 1° spatial resolution, 3 h output of mean significant wave height (average of 1/3 largest wave heights), dominant period (time between two consecutive wave crests or troughs), and direction (Dp; degrees north from which the waves are traveling) from January 1997 – March 2010 was used. Kingman and Palmyra were divided into 16 discrete 22.5° sectors (each 90° segment subdivided into four). Long-term climatological means in wave energy were calculated by averaging all 3 h time steps over the entire time series for each sector. Because wave energy standard deviation was highly correlated with the mean (r = 93.3%), it was not included as an independent predictor in the analyses. The WWIII model output reflects deep-water wave energy, not a direct measurement of wave stress across different habitats and depths. Because previous research has shown a strong linear relationship between deep-water offshore waves and wave-induced currents on reef-ecosystems (Hearn, 1999), this method is a good first-order approximation of wave forcing on coral reefs. To test for horizontal changes in wave energy, not changes as a direct result of changes in depth or habitat type, we limit comparisons of wave energy to forereef benthic communities at 10 m.

Statistics

Patterns of benthic communities were investigated at three taxonomic resolutions: (1) percent cover of calcified versus non-calcified (fleshy) organisms, (2) percent cover of major functional groups (hard coral, soft coral, CCA, other calcified macroalgae, fleshy macroalgae, other), and (3) percent cover of genera. All analyses were performed using R 2.15.1 (R Development Core Team, http://www.r-project.org) unless otherwise stated. We used a permutational multivariate analysis of variance (PERMANOVA) (Anderson, 2001) using the adonis function (vegan package) to test the effect of reef habitat and depth, and all subsequent pairwise comparisons within each factor. Results of each PERMANOVA were visualized with a canonical analysis of principal coordinates (CAP) based on a discriminant analysis (Anderson & Willis, 2003) using the CAPdiscrim function (BiodiversityR package). Individual variables that might be responsible for any group differences in the CAP analysis were investigated by calculating Spearman’s Rank correlations of the canonical ordination axes with the original variables. Variables with strong correlations (in this study, ≥0.4) were identified as “indicator genera” (i.e., driving group separation in multivariate space). Indicator genera need not be the most dominant (i.e. the organisms with the highest overall percent cover), but instead are those organisms contributing most to within-group similarity, while simultaneously contributing most to between-group dissimilarity. We provide the “allocation success” results for each CAP analysis in Table S2. Allocation success (expressed as a percentage) gives a measure of how distinct a group is relative to all other groups (with group defined as a level within a factor; e.g., the level forereef within the factor reef habitat). Allocation success indicated a more distinct group than expected by chance alone when values exceeded 100/n, with n being the number of a priori defined groups. All PERMANOVA and CAP analyses were based on 10 000 random permutations of the raw data.

To quantify the proportion of variation in benthic communities explained by horizontal gradients in wave energy, we used a permutational distance-based multivariate linear model (McArdle & Anderson, 2001) using the Fortran program DisTLM_foward (Anderson, 2003). To better explore the relationships between benthic communities and wave energy on the forereef at each atoll, we calculated univariate regression models for major functional groups and individual indicator genera that dominated the benthos. These univariate models should be considered more like data exploration, unlike the formal hypothesis testing of the multivariate analyses.

Finally, the spatial clustering of benthic taxa was calculated using dispersion-based weighting (Clarke et al., 2006), which measures deviations of the response variable (e.g., hard coral cover) from a generalized Poisson distribution using a test by permutation (1000 random permutations of the raw data). This approach is robust to flexible rules of clustering behaviour, as may be exhibited by hard coral communities, and for data not displaying a normal distribution, such as the percent cover data used in this study (Clarke et al., 2006). The clustering measure is given by the D statistic (variance to mean ratio), with higher values representing higher levels of spatial clustering (Clarke et al., 2006). Values of D were determined at the site level, with individual quadrats within any given site acting as the units of replication.

Results

Reef habitats at Kingman and Palmyra

Kingman’s lagoon was generally deep (>30 m) and contained numerous patch reefs 50–200 m in diameter, extending 2–10 m from the surface (Figs. 2A–2D and 4A). Kingman’s backreef slopes were steep (30–50° inclination) and extended beyond 30 m depth in many places before merging with the lagoon floor (Figs. 2E–2F and 4A). Kingman’s forereefs in the north and south gradually sloped for approximately 30–60 m out from the reef crest before dropping off sharply beyond 20 m depth (Figs. 2G–2H and 4A).

Figure 2 Reef habitats at Kingman Reef.

(A, B) Lagoonal patch reef dominated by massive and branching Porites. The shallow areas (<5 m) of the patch reefs are dominated by mushroom corals (Fungia) and giant clams (Tridacna) (C, D). The exposed shallow (<5 m) (E) and the deeper backreef (10 m) (F) have a steep slope incline (30–50°). Kingman’s forereef habitat, where plating and branching Acropora corals dominate at 10 m (G) and massive Porites characterize the deeper depths (H). Photo credits: ES (A–C, E, G, H), JES (D), GJW (F).

Figure 3 Reef habitats at Palmyra Atoll.

Palmyra’s shallow (<5 m) western reef terrace (A–B) gradually slopes to merge with a deep terrace habitat dominated by massive Porites and soft corals at 20 m (C). Palmyra’s forereef habitats generally have a steep incline and are dominated by corymbose Pocillopora corals and massive Porites (D–F). The only true backreef at Palmyra is located in the northeast (G). This shallow habitat (<3 m) boasts the highest cover of hard coral on the atoll and is in stark contrast to the heavily degraded lagoon habitat (H). Photo credits: Franklin Viola (A, C, G), GJW (B, H), Zafer Kizilkaya (D–F).

Figure 4 Stylized reef profiles from the two atolls.

Reef profiles for Kingman and Palmyra showing changes in percent cover of major benthic functional groups for: the transition from the lagoonal patch reef, across the backreef, and across forereef depths at Kingman (A), the transition from the shallow to the deeper reef terrace at Palmyra (B), and across forereef depths and the unique northeast backreef at Palmyra (C). Dominant hard coral genera (two most abundant in terms of percent cover in rank order) are indicated across habitats and depths in italics. Dashed lines indicate the position and direction of the cross-sectional profile.

Palmyra’s reef habitats included elongated reef terraces in the west and east. The shallow (<5 m) portion of the western terrace (Figs. 3A–3B) merged with lagoon flats to the east and gradually sloped to the west for approximately 4–5 km before becoming a deeper (>20 m) sloping terrace habitat (Fig. 3C) and dropping off sharply beyond 30 m depth (Fig. 4B). In the far east, the shallow terrace habitat was extremely exposed, and quantitative surveys were not possible. Qualitatively, the shallow eastern terrace was dominated by sand, with sparse patches of coral. Within approximately 5 km east of this sand zone, the eastern terrace reached depths of 20 m before dropping off sharply beyond 30 m depth. Forereef habitats along the north and south coasts of Palmyra (Figs. 3D–3F) merged with the deeper terrace habitats at their western and eastern extremities. Forereef steepness varied, but generally a sharp dropoff occurred at 30 m depth (Fig. 4C). Shallow (<1 m) lagoon flats ran along the north and south shores of Palmyra, often merging with the reef crest (Fig. 4C); thus, true backreef habitats were restricted to the northeast (Figs. 1C and 3G) and were not comparable to Kingman’s backreef.

Overall functional group cover

At Kingman, across all reef habitats and depths, mean hard coral cover was 42%, crustose coralline algae (CCA) 22%, other calcified macroalgae 8%, soft coral 6%, fleshy macroalgae 0.7%, and turf algae 12%. At Palmyra, excluding the northeast backreef habitat, mean hard coral cover was 29%, CCA 24%, other calcified macroalgae 14%, soft coral 7%, fleshy macroalgae 5%, and turf algae 19%. Mean hard coral cover at Palmyra’s northeast backreef was 76%, the highest of any habitat surveyed at the two atolls (Fig. 4C). The mean cover of calcified organisms (sum of hard coral, CCA, other calcified macroalgae, and other calcified invertebrates) was 74% at Kingman (range = 54–86%) and 67% at Palmyra (range = 56–77%) (Table S1).

Benthic community patterns across reef habitats

At Kingman, benthic communities differed across reef habitats at 10 m at all three taxonomic resolutions (Table 1, Table S2). The patch reef and forereef were characterized by a higher cover of calcified organisms, particularly hard coral, than the backreef, which had a higher cover of turf algae (Fig. 4A). At the genus level, the patch reef and backreef were dominated by Porites, which composed ≥80% of the hard coral cover present in both habitats (Table 2). In contrast, on the forereef, the coral communities were more diverse, with a greater number of genera contributing to overall hard coral cover. Acropora and Pocillopora dominated the forereef habitat and together with Porites and Montipora composed >80% of the hard coral cover present (Table 2).

Table 1 Summary statistics of within-island variations in benthic communities at three taxonomic resolutions at Kingman Reef and Palmyra Atoll.

Island	Taxonomic resolution	Factor	Depth strata (m)	Pseudo-F	P-value	Pairwise comparisons	
Kingman	Calcified vs. non-calcified	Reef habitat	10	6.54392,17	0.0005	P F | B	
		Forereef depth strata	5, 10, 20	5.13522,16	0.022	10 | 20	
	Functional group	Reef habitat	10	8.04822,17	0.0001	P | B | F	
		Forereef depth strata	5, 10, 20	8.89702,16	0.0001	5 | 10 | 20	
	Genus	Reef habitat	10	10.5682,17	0.0001	P | B | F	
		Forereef depth strata	5, 10, 20	5.74452,16	0.0001	5 | 10 | 20	
Palmyra	Calcified vs. non-calcified	Terrace depth strata	5, 20	11.9431,13	0.0043	NA	
		Forereef depth strata	5, 10, 20	1.09182,31	0.3508	ns	
	Functional group	Terrace depth strata	5, 20	8.28581,13	0.0011	NA	
		Forereef depth strata	5, 10, 20	2.76782,31	0.0061	5 | 20	
	Genus	Terrace depth strata	5, 20	14.0571,13	0.0011	NA	
		Forereef depth strata	5, 10, 20	2.57562,31	0.0024	5 10 | 20	
Notes.

Results of permutational multivariate analysis of variance analyses; | indicates significant differences between groups in pairwise comparisons; ns, non-significant; NA, pairwise comparison not applicable due to only two groups.

Degrees of freedom for each test are shown as subscripts for each respective Pseudo-F value.

Table 2 Abundance (mean percent cover) of the top 10 hard corals across reef habitats (patch reef, backreef and forereef) and forereef depths (5, 10, 20 m) at Kingman Reef.

The relative contribution of each genus to overall hard coral cover is shown (Rel). Corals in bold make up ≥50% of overall hard coral cover. For complete cover values for all hard coral genera see Table S1.

						Forereef	
Patch Reef - 10 m	Backreef - 10 m	5 m		10 m		20 m		
Genus	Mean cover	Rel	Genus	Mean cover	Rel	Genus	Mean cover	Rel	Genus	Mean cover	Rel	Genus	Mean cover	Rel	
Porites	50.6	88.5	Porites	20.7	79.6	Acropora	13.7	37.1	Acropora	25.2	44.8	Porites	10.7	33.0	
Fungia	3.2	5.6	Fungia	1.3	5.0	Pocillopora	8.0	21.7	Pocillopora	9.5	16.9	Favia	6.0	18.5	
Favia	1.3	2.3	Acropora	1.2	4.6	Porites	5.0	13.6	Porites	7.3	13.0	Pocillopora	3.8	11.7	
Pocillopora	0.9	1.6	Favia	1.0	3.8	Montipora	3.7	10.0	Montipora	5.7	10.1	Acropora	3.0	9.3	
Pavona	0.5	0.9	Turbinaria	0.9	3.5	Pavona	2.1	5.7	Pavona	2.3	4.1	Lobophyllia	2.5	7.7	
Montipora	0.3	0.5	Montipora	0.4	1.5	Favia	1.2	3.3	Favia	2.3	4.1	Pavona	1.1	3.4	
Stylophora	0.2	0.3	Herpolitha	0.1	0.4	Leptastrea	1.0	2.7	Hydnophora	1.3	2.3	Montipora	0.9	2.8	
Astreopora	0.03	0.1	Pavona	0.1	0.4	Montastrea	1.0	2.7	Stylophora	0.6	1.1	Platygyra	0.8	2.5	
Platygyra	0.03	0.1	Favites	0.09	0.3	Favites	0.5	1.4	Favites	0.6	1.1	Fungia	0.7	2.2	
Montastrea	0.02	0.0	Astreopora	0.04	0.2	Hydnophora	0.3	0.8	Platygyra	0.3	0.5	Leptastrea	0.5	1.5	

Indicator genera driving separation among Kingman’s reef habitats included a range of functional groups (Fig. 5A). On patch reefs, the coral Porites both dominated in percent cover and was the strongest indictor genera of this habitat (Fig. 5A). Other indicator genera on patch reefs were Fungia corals, the giant clam Tridacna, and the fleshy macroalga Avrainvillea (Fig. 5A). While the overall cover of Avrainvillea was low (Table S1), the rare nature of this alga made it a particularly strong discriminating species between reef habitats. On the backreef, although Porites was the most dominant coral, it was not a strong indicator distinguishing this habitat. Indicative genera on the backreef were turf algae, sand, the calcifying macroalga Halimeda, and the hard coral Turbinaria (Fig. 5A). Finally, on the forereef, three of the four most abundant corals (Acropora, Pocillopora, and Montipora) were also strong indicator genera for this habitat (Fig. 5A). Other indicator genera on the forereef included the calcifying macroalga Peyssonnelia, the fleshy macroalga Dictyosphaeria, leather corals, and the hard corals Pavona, Montastrea, Lobophyllia, Favites, and Hydnophora (Fig. 5A).

Figure 5 Canonical plots.

Canonical analysis of principal coordinates (CAP) based on a discriminant analysis, showing those benthic genera responsible for separation across reef habitats at Kingman (A) and across forereef depths on Kingman (B) and Palmyra (C). The squared canonical correlation value for the first two ordination axes is shown in parentheses in each case. Vector lines represent Spearman’s Rank correlations (threshold set at ≤0.4). The length of each vector line is proportional to the strength of the correlation. CCA, crustose coralline algae; Pachy/Stereo, Pachyclavularia/Stereonephthya.

Benthic community patterns across depths

At Kingman, benthic communities differed across forereef depths, particularly at the functional group and genus level (Table 1, Table S2). Hard coral cover peaked at mid depths (10 m), CCA decreased with depth, other calcifying macroalgae increased with depth, and soft coral and turf algae peaked at deeper (20 m) depths (Fig. 4A). At 5 and 10 m, Acropora and Pocillopora dominated, representing >50% of hard coral cover at these depths (Table 2). At 20 m, Porites and Favia were the dominant corals (Table 2).

Indicator taxa driving separation among forereef depths on Kingman included a range of functional groups (Fig. 5B). In addition to high CCA cover, shallow depths were characterized by the giant clam Tridacna, the hard coral Pavona, and the hydrozoan fire coral Millepora. The corals Acropora and Pocillopora both had high percent cover values at these depths, and together with Montipora, were strong indicators for shallow and mid-depths (Fig. 5B). At 20 m, the increase in calcified macroalgae cover was strongly driven by Halimeda, and the increase in soft coral cover was driven by leather corals and Pachyclavularia/Stereonephthya (Fig. 5B). Turf algae were also characteristic of the 20 m depth strata. Characteristic hard corals at 20 m, in addition to the two most abundant (Porites and Favia), were Fungia, Lobophyllia, Platygyra, and Herpolitha (Fig. 5B).

On Palmyra’s reef terrace, benthic communities differed between shallow and deeper depths (Table 1, Table S2). Hard coral, CCA, and fleshy macroalgae (particularly Lobophora) characterized the shallows, while soft corals and other calcified macroalgae, particularly Peyssonnelia, were indicative of deeper depths (Fig. 4B, Table S1). At 5 m, Montipora dominated and represented 71% of the hard coral cover present. Montipora and Acropora together composed >80% of the hard coral cover on the shallow terrace (Table 3). At 20 m, Porites dominated and represented 67% of the hard coral cover present. In conjunction with Pocillopora and Turbinaria, these three genera composed >80% of the hard coral cover present on Palmyra’s deep terrace (Table 3).

Table 3 Abundance (mean percent cover) of the top 10 hard corals across reef habitats (terrace and forereef) and depths (5, 10, 20 m) at Palmyra Atoll.

The relative contribution of each genus to overall hard coral cover is shown (Rel). Genera in bold make up ≥50% of overall hard coral cover. For complete cover values for all hard coral genera see Table S1.

Terrace	Forereef	
5 m		20 m		5 m		10 m		20 m		
Genus	Mean cover	Rel	Genus	Mean cover	Rel	Genus	Mean cover	Rel	Genus	Mean cover	Rel	Genus	Mean cover	Rel	
Montipora	34.4	70.9	Porites	21.0	67.3	Pocillopora	6.2	30.8	Pocillopora	6.3	25.9	Porites	7.2	32.0	
Acropora	6.7	13.8	Pocillopora	2.6	8.4	Porites	3.8	18.9	Porites	5.0	20.6	Pocillopora	2.8	12.4	
Psammocora	2.8	5.8	Turbinaria	2.5	8.0	Montipora	2.4	11.9	Montipora	3.2	13.2	Pavona	2.2	9.8	
Pocillopora	2.4	4.9	Montipora	1.8	5.8	Pavona	1.7	8.5	Pavona	2.5	10.3	Favia	2.1	9.3	
Astreopora	1.0	2.1	Favia	1.1	3.5	Favia	1.3	6.5	Favia	2.3	9.5	Turbinaria	1.9	8.4	
Pavona	0.7	1.4	Pavona	0.7	2.2	Favites	1.1	5.5	Hydnophora	1.0	4.1	Montipora	1.6	7.1	
Favia	0.3	0.6	Favites	0.5	1.6	Acropora	1.1	5.5	Lobophyllia	0.9	3.7	Hydnophora	1.0	4.4	
Porites	0.1	0.2	Montastrea	0.3	1.0	Montastrea	1.0	5.0	Turbinaria	0.7	2.9	Favites	0.7	3.1	
Stylophora	0.08	0.2	Leptastrea	0.2	0.6	Leptastrea	0.6	3.0	Montastrea	0.6	2.5	Fungia	0.5	2.2	
Fungia	0.07	0.1	Acropora	0.2	0.6	Turbinaria	0.4	2.0	Favites	0.5	2.1	Acropora	0.5	2.2	

On Palmyra’s forereef, the overall cover of calcified and non-calcified organisms did not vary across depths (Table 1), but the communities comprising each of the two groups changed. Hard coral and turf algae cover was similar across depths andthe cover of fleshy macroalgae peaked in the shallows. CCA cover progressively decreased with depth, while the cover of other calcified macroalgae (particularly Peyssonnelia) and soft corals increased with depth (Fig. 4C). The hard coral community was generally dominated by Pocillopora and Porites, which together with Montipora and Pavona composed >50% of the hard coral cover at each forereef depth (Table 3). The hard coral community on the forereef was more diverse than on the reef terrace, with six to seven genera comprising 80% of the hard coral cover across forereef depths (Table 3).

The hard corals Pocillopora and Montastrea, together with the fleshy macroalgae Caulerpa and Lobophora, were indicative of shallow to mid depths, with Lobophora particularly indicative of shallow depths (Fig. 5C). The most abundant hard corals at 20 m (Porites, Pocillopora, and Pavona) were not strong indicators for this deeper depth. Instead, the hard corals Psammocora and Leptoseris, and corals within the family Fungiidae (Fungia, Herpolitha, and Halomitra), were characteristic of 20 m depths on Palmyra’s forereef (Fig. 5C).

Horizontal wave energy gradients

Kingman and Palmyra experience similar wave forcing conditions, with high wave energy (40– 60 kW/m) from the north and northwest (292.5–360°); moderate energy (20– 30 kW/m) from the south, west, and northeast (180–292.5° and 0–90°); and low energy (< 20 kW/m) from the southeast (90–157.5°) (Fig. 6A). Over the 13 + year data record, 62% of all incident waves were within the 45–90° sectors (largest sector value of 32.1%; Fig. 6B). Waves from the north and northwest accounted for 18% of all incoming waves (largest sector value of 12.2%), while all other sectors accounted for <20% (largest sector value of 6.3%).

Figure 6 Wave energy plots for Kingman and Palmyra.

Mean wave power at Kingman and Palmyra in kW/m calculated for 16 discrete sectors, each spanning 22.5° starting from 0°. Power calculated using significant wave height, dominant period and direction from NOAA’s Wave Watch III, a 3 h interval, 1° spatial resolution wave model from January 1997 – March 2010 (A). Numbers in italics represent wave energy values in kW/m. Wave event frequency (B) is the number of wave events (3 h output values) in each 22.5° sector divided by the total number of events over the 13 + year time period. Values in italics represent the fraction of waves.

At 10 m on Kingman’s forereef, horizontal gradients in wave energy explained 51% of the variation in percent cover of calcified versus non-calcified organisms (distlm, Pseudo-F = 13.484, P = 0.0073). Percent cover of calcified organisms decreased as wave energy increased, with an approximately 20% drop in cover with a doubling of wave energy (Fig. 7A). In contrast, cover of non-calcified organisms increased approximately threefold with a doubling of wave energy (Fig. 7A). At the functional group level, 48% of the variation in benthic community cover was explained by variation in wave energy (distlm, Pseudo-F = 12.020, P = 0.0007). Hard coral cover decreased from approximately 60% to 20% with a doubling in wave energy (Fig. 7B); this appeared to be driving the negative relationship between calcified cover and wave energy. In contrast, turf cover increased approximately threefold with a doubling in wave energy (Fig. 7B); this appeared to be driving the positive relationship between non-calcified cover and wave energy. Even at the genus level, variations in wave energy explained 40% of the variation in benthic community cover (distlm, Pseudo-F = 8.482, P = 0.0001). Specifically, cover of the hard coral Acropora was negatively related to wave energy, decreasing from approximately 30% to almost zero with a doubling of wave energy (Fig. 7C). In contrast, although a partially calcified organism, cover of the encrusting red alga Peyssonnelia was positively related to wave energy, doubling with a doubling of wave energy (Fig. 7C). Finally, the spatial clustering of hard coral cover at 10 m approximately doubled as wave energy doubled (Fig. 7D). No one genus appeared predominantly responsible for this relationship. No other functional groups or genus-level indicator variables showed strong relationships with wave energy.

Figure 7 Relationships between benthos and wave energy.

Relationship between benthic communities and wave power (kW/m) on Kingman’s forereef at 10 m.

On Palmyra’s forereef at 10 m, there were no significant relationships between horizontal gradients in wave energy and percent cover of calcified versus non-calcified organisms (distlm, Pseudo-F = 0.2802, P = 0.591), cover of various functional groups (distlm, Pseudo-F = 1.2121, P = 0.329), or cover of various genera (distlm, Pseudo-F = 1.2816, P = 0.237).

Discussion

Benthic communities at Kingman Reef and Palmyra Atoll were dominated by reef-building calcifying organisms, namely hard corals and crustose coralline algae. This is consistent with patterns observed at other remote, often uninhabited, islands in the Pacific, including the US Phoenix Islands (Vroom et al., 2010) and parts of the Northwestern Hawaiian Islands (Vroom & Braun, 2010; Page-Albins et al., 2012). In contrast, more impacted reef communities, such as the populated islands of Kiritimati and Tabuaeran that neighbor Kingman and Palmyra, are typically characterized by a higher cover of fleshy turf and macroalgae (Sandin et al., 2008).

Reef Habitat

At Kingman and Palmyra, although differences in benthic communities were observed among reef habitats, some consistent patterns emerged. Hard coral cover was generally high, and one or two genera heavily dominated in sheltered habitats such as Kingman’s patch reefs and Palmyra’s shallow reef terrace. The intermediate disturbance hypothesis predicts that, in the absence of frequent disturbance, a climax community emerges that becomes dominated by a small number of competitively superior species (Connell, 1978; Rogers, 1993). Kingman’s patch reefs were dominated almost exclusively by slow-growing, massive Porites spp. corals, which are not rapid colonists following disturbance events (Glynn, 1993). This provides indirect evidence of competitive exclusion and suggests that Kingman’s patch reefs represent a low disturbance environment with mature communities at the equilibrium end of the scale within a nonequilibrium system (Hughes & Connell, 1999). Furthermore, as the establishment of climax communities is often prevented by routine mortality (Tanner, Hughes & Connell, 1994), the mortality of these massive Porites must be low enough to maintain a monopolization of space. This type of monospecific dominance in sheltered reef habitats by slow-growing Porites has been noted elsewhere in the Pacific, for example in parts of the Hawaiian Archipelago (Grigg & Maragos, 1974; Grigg, 1983; Page-Albins et al., 2012). Evidence suggests that Palmyra’s shallow reef terrace was once heavily dominated by the branching coral Acropora acuminata (Williams et al., 2010), likely as a result of its fast growth rate, its ability to spread via fragmentation, and its ability to shade competitors (Lang & Chornesky, 1990). The 1998 mass bleaching event caused high mortality in A. acuminata (Williams et al., 2010), and Palmyra’s shallow terrace is now dominated by encrusting Montipora; however, Acropora is the second most abundant coral genus. While Palmyra’s shallow terrace has high hard coral cover and relatively low diversity, this habitat is still unlikely in an equilibrium state. A recent disease outbreak targeting tabular Acropora on Palmyra’s shallow reef terrace (Williams et al., 2011a) highlights the dynamic nature of this coral community, with routine mortality seemingly preventing monopolization of space by fast-growing Acropora corals.

Kingman’s forereef, while having similar hard coral cover to the patch reefs, had a higher diversity of coral genera. Here, fast-growing branching and tabular Acropora and corymbose Pocillopora dominated, not slow-growing massive Porites. The high diversity and abundance of fast-growing corals on Kingman’s forereef suggests a nonequilibrium community, probably in an earlier successional state (Hughes & Connell, 1999). Oceanic forereef habitats are generally more exposed to physical disturbance events, such as large swells, than more sheltered patch reef habitats (Storlazzi et al., 2005). Frequent disturbance events (relative to the life histories of the local assemblages) on Kingman’s forereef may be keeping benthic communities in earlier successional stages (Grigg, 1983) and preventing competitive dominance by few species (Connell, 1978; Rogers, 1993).

Finally, Kingman’s backreef had lower coral cover and higher turf algae cover than both the patch reefs and forereef, but as with the patch reefs, coral cover was heavily dominated by massive Porites. On the backreef, conditions may be stable enough to allow competitive dominance by Porites but unstable enough to prevent high Porites cover from establishing (Hughes & Connell, 1999). One possible disturbance mechanism could be the transport of debris, such as live and dead coral fragments, by large swell events from the forereef over the reef crest and onto the backreef slope (pers. obs.). Due to the steepness of Kingman’s backreef slope, debris could easily reach a depth of 10 m and still cause physical disturbance. The opening of space on a more regular basis by debris would help to explain the higher cover of turf algae, a rapid early colonist of bare space following disturbance (Grigg, 1983). It is also likely that grazing from the abundant herbivorous fishes at Kingman (Friedlander et al., 2010) prevents transition of this turf algae toward macroalgal dominance on the backreef.

Depth

Consistent patterns of benthic community zonation emerged across the three forereef depths at Kingman and Palmyra (5, 10, 20 m), and these patterns largely held true between the two depths on Palmyra’s reef terrace as well (5, 20 m). With increased depth, the most consistent patterns were: a decrease in crustose coralline algae (CCA) and fleshy macroalgae, an increase in other calcified macroalgae, and an increase in soft coral cover. Patterns of hard coral cover were less consistent, remaining similar across depths on Palmyra’s forereef, higher in the shallows than the deeper reef on Palmyra’s terrace, and peaking at intermediate depths on Kingman’s forereef.

A peak in hard coral cover at intermediate depths appears to be a common feature of forereef habitats (Huston, 1985). At shallow depths, harsh environmental conditions, such as high oscillatory flow and irradiance, can inhibit coral feeding and cause photo damage (Sebens & Johnson, 1991; Brown, 1997). Such physical conditions can prevent the establishment of benthic communities with high coral cover (Rogers, 1993; Hughes & Connell, 1999). Meanwhile, lower irradiance at deeper depths can limit photosynthesis by zooxanthellae, while very low flow can lead to the establishment of boundary layers around the coral surface and inhibit nutrient uptake, thus depressing coral respiration and growth (Patterson, Sebens & Olson, 1991). Mid-depths on Kingman’s forereef thus may represent an energetic optimum for hard corals, increasing overall cover. At mid-depths, levels of irradiance and water flow may maximize the balance between energy intake and respiration (Sebens, 1984) while providing adequate settlement substrate, such as CCA, for coral recruitment (Price, 2010). This interpretation should be applied cautiously to other locations, however, as not all reef environments share the same patterns of wave energy and irradiance.

The increase of soft coral cover with depth on Kingman’s and Palmyra’s forereefs may reflect the fact that zooxanthellate soft corals rely only moderately on phototrophy to supply their carbon requirements. Soft corals often gain a large proportion of their food via heterotrophy, often more effectively than scleractinian corals (Fabricius & Dommisse, 2000). Therefore, for soft corals, proximity to allochthonous particulate food from upwelling (i.e., at increased depths) is likely more beneficial than proximity to higher irradiance (i.e., shallower depths), which is often accompanied by high wave stress. Thus, it is hypothesized that higher soft coral cover at depth at Kingman and Palmyra occurs because of higher food availability. The encrusting nature of CCA and its ability to photoacclimate makes it particularly suited to shallow reef slope environments where there is increased oscillatory flow and irradiance (Sheppard, 1980; Bulleri, 2006). CCA cover decreased with depth on Kingman’s forereef and Palmyra’s forereef and terrace. This likely reflects a shift from thicker-crusted species in the shallows that are more competitive with other encrusting macroalgae, such as Lobophora and Peyssonnelia, and that are more resistant to high flow and sediment abrasion, to species with thinner crusts at deeper depths (Dethier, Paull & Woodbury, 1991). The biomass of herbivorous fishes is higher at shallower depths on Kingman’s forereef (Friedlander et al., 2010), potentially leading to increased grazing pressure on algal turfs and indirectly promoting and facilitating CCA growth and persistence in the shallows (Smith, Hunter & Smith, 2010).

In contrast to CCA, percent cover of the calcified macroalgae Halimeda and Peyssonnelia increased with depth. Many genera of calcified macroalgae, such as Halimeda, contain specialized photosynthetic accessory pigments such as siphonoxanthin and siphonein that allow them to effectively harvest light in deeper environments (Drew, 2011). In contrast, due to its upright structure, Halimeda is likely vulnerable to being dislodged by wave action in the shallows (Dethier, Paull & Woodbury, 1991). Increased abundance of herbivorous fishes in the shallows may also increase grazing there (Friedlander et al., 2010), reducing the cover and changing the composition of algal assemblages. The top-down effects of herbivory on the emergent structure of the algal assemblage are difficult to predict, however, without detailed knowledge of foraging patterns and functional responses (e.g., is herbivory under-, over-, or perfectly compensatory with changes in algal production?) (Gruner et al., 2008). The increase of Halimeda and Peyssonnelia with depth could also reflect increased supply of nutrient-rich water from upwelling or internal tides/waves, promoting growth (Smith et al., 2004), as well as competitive release from CCA (Dethier, Paull & Woodbury, 1991). CCA abundance may be independent of nutrient supply (Belliveau & Paul, 2002) or more dependent on high flux rates to reduce boundary layer thickness and ensure that nutrients actually reach the thallus surface.

Kingman had virtually no fleshy macroalgae at any depth. The cover of the most dominant fleshy macroalga at Palmyra, Lobophora, was highest in the shallows, likely due to its low-lying, fan-like morphology, affording it protection from high oscillatory flow. The different relationships of macroalgal taxa with depth emphasize the importance of discriminating between calcified and fleshy macroalgae when linking macroalgal cover to environmental conditions on coral reefs.

Wave energy

Forereef benthic community organization at 10 m depths at Kingman was consistent with predictions of a simple univariate model of wave energy, such that the cover of more wave-tolerant organisms increased with increased wave energy. This was not true for Palmyra, where wave energy showed no clear patterns with benthic community organization.

On Kingman’s forereef, the negative relationship between hard coral cover, particularly Acropora, and wave energy at shallow and mid-depths is most likely due to physical disturbance by wave action. The dominant Acropora morphologies on Kingman’s forereef are branching and tabular (Kenyon, Maragos & Wilkinson, 2010), both of which have high colony shape factors and are thus vulnerable to being dislodged from the substrate or damaged from waterborne projectiles (Madin, 2005). In a community dominated by acroporid corals, then, it follows that increased wave energy results in decreased overall hard coral cover. In contrast, cover of turf algae and the calcified encrusting red alga Peyssonnelia both increased with increased wave energy. Following an inhibition model of succession (Connell & Slatyer, 1977), these low-lying, wave-tolerant organisms may become competitively superior under high wave energy conditions (Page-Albins et al., 2012). Alternatively, the higher cover of these fast-growing, early successional organisms could reflect a community continually re-set to an earlier successional state by repeated physical disturbances (Rogers, 1993; Hughes & Connell, 1999). Analyses of time-series data from permanent plots would be required to discern which of these mechanisms is predominantly producing the observed benthic community patterns.

There may be several reasons why wave energy related strongly to mid-depth forereef benthic communities at Kingman but not at Palmyra, two reefs only 60 km apart that experience similar wave energy climatologies. First, Kingman is dominated by fast-growing, fragile morphologies of Acropora corals, while Palmyra is dominated by fast-growing corymbose Pocillopora corals and slow-growing massive Porites, both of which are generally more wave-tolerant than Acropora. Second, in a previous study at Palmyra, sedimentation (importantly, the percentage of very fine sediment) was strongly related to overall hard coral cover and genus-level benthic community patterns (Williams et al., 2011b). The degraded lagoon at Palmyra contains high levels of fine sediment that is occasionally transported across the reef during the change in tidal state (pers. obs.). The reef sites most impacted by sedimentation are associated with lower hard coral cover (Williams et al., 2011b) and a higher prevalence of coral bleaching during thermal stress events (Williams et al., 2010). This strong relationship between sedimentation and benthic communities at Palmyra may thus be obscuring any easily detectable signal of wave energy on community organization. Finally, waves can be highly complex due to refraction, dissipation and other wave-bathymetry interactions. The use of deep water wave information in discrete sectors may underrepresent the spatial complexity of nearshore wave forcing, potentially contributing to the observed decoupling of wave energy and benthic community composition around Palmyra. We have not accounted for other physical oceanographic forcings that may be pertinent to benthic reef communities, such as internal tides and patterns of lagoon outflow. A nearshore hydrodynamic model would presumably capture additional physical forcings and therefore be a more appropriate tool for exploring biophysical coupling at Palmyra.

Kingman Reef and Palmyra Atoll provide an opportunity to study benthic dynamics and biophysical coupling where calcifying organisms, such as hard corals and CCA, dominate the benthos, where local human impacts are absent, and where the remote oceanic nature of these reef systems provides unique biophysical settings. Elucidating the driving forces of ecological dynamics for such systems will allow discrepancies with more impacted reefs in similar oceanographic settings to be deciphered. Such information should provide a context specific understanding of the relative importance of local human impacts versus natural variations in oceanography and climate on coral reef community organization and dynamics.

Supplemental Information

Supplemental Information Supplemental Information

Benthic percent cover values identified to the lowest taxonomic resolution possible across reef habitats and depths at Kingman Reef and Palmyra Atoll and subsequent allocation success values for formal testing of group differences.

Click here for additional data file.

All activities at Kingman Reef and Palmyra Atoll were conducted under the US Fish and Wildlife Service special use permits 12533-07021 and 12533-10010. We additionally thank the US Fish and Wildlife Service and The Nature Conservancy for logistical support. We thank Rachel Morrison, Franklin Viola, Zafer Kizilkaya, Lisa Wedding, and Nichole Price for contributions to the manuscript. Finally, we thank Peter Vroom, Abel Valdivia and a third anonymous reviewer for comments that greatly improved this manuscript. SIO is a member of the Palmyra Atoll Research Consortium (PARC) and the Reefs Tomorrow Initiative. This is PARC publication number PARC-0095.

Additional Information and Declarations

Competing Interests

Author Contributions

Field Study Permissions

We have no competing interests.

Gareth J. Williams and Jennifer E. Smith conceived and designed the experiments, performed the experiments, analyzed the data, wrote the paper.

Eric J. Conklin performed the experiments.

Jamison M. Gove analyzed the data, wrote the paper.

Enric Sala contributed reagents/materials/analysis tools.

Stuart A. Sandin performed the experiments, wrote the paper.

The following information was supplied relating to ethical approvals (i.e. approving body and any reference numbers):

All activities at Kingman Reef and Palmyra Atoll were conducted under the US Fish and Wildlife Service special use permits 12533-07021 and 12533-10010.

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
