# Peer review of "Benthic communities at two remote Pacific coral reefs: effects of reef habitat, depth, and wave energy gradients on spatial patterns"

_PeerJ, doi:10.7717/peerj.81_

## Round 0.1 · original submission · Minor Revisions

Can you please include an image illustrating what is meant by "turf" in this manuscript?

·

Basic reporting

Page 2, lines 33-37; page 3, lines 48-50: Page et al. (2012) recently published a manuscript describing patterns in benthic community structure along a wave exposure gradient at Pearl and Hermes Atoll. This manuscript should probably also be cited. “Page-Albins, Kimberly N., Peter S. Vroom, Ronald Hoeke, Mark A. Albins, Celia M. Smith (2012) Patterns in benthic coral reef communities at Pearl and Hermes Atoll along a wave exposure gradient. Pacific Science 66: 481-496.”

Pages 3-4, lines 64-65: Cite “Kenyon et al. (2012) Monitoring supports establishment of Pacific Remote Islands Marine National Monument. Proceedings of the 12th International Coral Reef Symposium, Cairns, Australia, 9-13 July 2012. 18F”

Page 5, line 108: A qualified is needed after “10 m”. For instance: “…, surveys were conducted at 10 m depths across all three reef habitats…”

Results section: The word “comprised” is used incorrectly throughout the manuscript. I think the word that is typically needed is “composed.” For example, page 12, line 256:

Examples: Incorrect: Montipora and Acropora together comprised > 80% of the hard coral cover.
Correct: Montipora and Acropora together composed > 80% of the hard coral cover.
Correct: Greater than 80% of the hard coral cover comprised Montipora and Acropora.

Page 10, line 218: I think Table 3 is cited in the text before Tables 1 and 2. Check order of Tables and Figures.

Page 11, line 221: In this one instance, a “≥” symbol is needed rather than “>” since Table 1 indicates that Porites equals 80% cover.

Page 13, line 263: Remove the word “cover” after hard coral. The sentence currently reads “Cover of hard coral cover and turf algae….”

Page 13, line 273: You’re only talking about one depth, so change the sentence to read: “… were not strong indicators for this deeper depth.”

Page 13, line 275: Change “deeper” to “20-m”

Page 20, line 424: A qualified is needed after “10 m”. For instance: “…community organization at 10 m depths at Kingman…”

Figure legends: sometimes figure sections (e.g., (a), (b)) are listed before the phrase describing the figure, and sometimes after. Be consistent. What is the style preferred by the journal?

Figure 5: I find the color symbols used to differentiate habitats in Figure 5a visually easier to quickly differentiate than the depths in Figures 5b, 5c. Maybe consider indicating different depths by colored symbols (different than the ones used to differentiate habitats in Figure 5a) rather than numbers.

Figure 5b: Millepora is spelled incorrectly (i.e., Millipora).

Table 1: The columns containing the relative contributions of coral genera don’t add up to 100%. Should there be an “other” category listed so that 100% of relative contribution is depicted?

Experimental design

The experimental design (i.e., photoquadrats taken along transect lines) is straight-forward, and an accepted method for determining percent cover of organisms in marine environments.

Page 5, line 105: You state that study sites were chosen at random, yet many appear almost equidistant from each other in Figure 1. Is random really the correct word here?

Page 5, lines 106-107: I think another phrase or sentence might be necessary at the end of this sentence. For instance; “Although reef flats are present on both atolls, their shallow and exposed nature made surveying them unsafe. Thus, no sampling occur in this habitat.”

Validity of the findings

Abstract: In addition to the possibility of increased nutrients at depth that may allow calcified algae such as Halimeda to flourish, many genera also have specialized photosynthetic accessory pigments such as siphonoxanthin and siphonein that allow them to effectively harvest light in deeper environments.

Page 3, lines 59-62: Although only relatively small amounts of algae might have been apparent in photoquads, Kingman supports a fairly diverse algal flora that should be referenced along with fish and corals. Cite: “Tsuda, Roy T., Jack R. Fisher, Peter S. Vroom (2012) Floristic account of the marine benthic algae from Jarvis Island and Kingman Reef, Line Islands, Central Pacific. Micronesica 43:14-50.”

Page 12, lines 243-251: You mention every organism shown in Fig. 5b except for Montipora at 10-m depths and turf algae at 20-m depths. Since everything else is listed in the text, I’d suggest also making reference to these additional two taxa.

Page 13, line 263: Fleshy macroalgae don’t appear to progressively decrease with depth in Figure 4c.

Page 15, lines 310-311: I disagree that fleshy macroalgae were virtually absent at Kingman and rare at Palmyra. They may have been visually absent in the photoquadrats analyzed, but a suite of macroalgae were likely present growing in holes and crevices, and under overhangs. Don’t forget that macroalgae are a necessary part of intact tropical reef ecosystems, and help support the base of the food chain. In healthy reef systems, they just don't usually grow out in the open because of grazing pressue.

Page 18, lines 392-393: I agree with the premise being presented that soft corals gain a large portion of food via heterotrophy and that upwelling at Kingman may increase the amount of food availability at depth. However, it seems too speculative to definitively state that soft coral cover is higher at 20 m for this reason. No actual experimentation has taken place at Kingman to prove that this is what is occurring. Maybe change the last sentence of the paragraph to something like: “Thus, it is hypothesized that higher soft coral cover at depth at Kingman and Palmyra occurs because of higher food availability.”

Page 20, line 417: Saying that virtually no fleshy macroalgae were present at any depth on Kingman kind-of contradicts Figure 5a that shows Avrainvillea and Dictyosphaeria as indicator genera driving separation among reef habitats.

Page 20, Wave Energy section. Also include findings from Page et al. (2012) in your Discussion.

Additional comments

I found the manuscript interesting and easy to read. It is an important contribution to the literature since it documents percent cover of a suite of organisms on two of the healthiest reef systems remaining on the planet. Good work!

·

Basic reporting

The manuscript “Benthic communities at two remote Pacific coral reefs: effects of reef habitat, depth, and wave energy gradients on spatial patterns” by Williams et al., is a generally well written and thorough document that delve into some of the physical-oceanography conditions that structure coral reef benthic communities on two of the most remote atolls of the world with virtually no human impacts. In particular, their main objective was to provide a detailed description of the benthic assemblage patterns across reef habitat, depth and wave exposure gradients. The principal findings support a large body of evidence and studies performed in the 1980s and 1990s about the natural environmental factors that control coral reefs. Below are some detailed comments that would improve the clarity of this manuscript.

Detailed comments:

Abstract
- Sentence 3: Specify that these percentages of benthic organism are percent cover (e.g., …Benthic communities “coverage” at both….)
- Sentence 5: Based on your results hard coral cover was the highest at the back reef of Palmyra instead of the patch reef of Kingman. Please specify if you are only explaining intra island comparisons.
- Sentence 11: At Palmyra hard coral cover pattern was inconsistent but at Kingman hard coral cover peaked at mid depths. Please specify that you are talking about Palmyra.
- In general make more clear when are you talking about Palmyra and/or Kingman in the abstract.

Introduction
- L38 and 39. Include references to support these statements.
- L43. Specify that this is for corals as a group including zooxanthellate and azooxanthellate corals, because hermatypic corals are constrained by depths due to their endosymbionts.
- L70. I suggest to start …”We first compare and contrast” … to better follow the subsequent sentences.

Materials and Method
- L112. Briefly explain why the effect of depth was not analyzed at 10m on the reef terrace at Palmyra as it was analyzed at Kingman atoll.
- L166. Briefly explain how “indicator variables” should be interpreted in driving major difference among reef habitats or depths, and how they can differ from dominant groups (e.g, groups with high percent cover might not be indicators of a given reef habitat)
- L178-179. Unclear sentence it should say ...should be considered more "like" data exploration…

Results
- L230-231. Stating that the backreef was “characterized” by turf algae, sand, etc, is a misleading statement. As stated before Porites was the dominant genus. Instead, state that these groups were indicators of difference among habitats based on the discriminant analysis.
- L244. Substitute “characterized” here as it mislead the reader. Use “indicative” when referring to groups that are good to differentiate among habitats.
- L268. Unclear statement. Are the six to seven genera compromising the 80% of the hard coral cover of reef terrace or forereef?
- L280-281. Label the panel of Fig.6 A and B and use this to call the figures in the text.
- L285. How much does wave energy explain for non-calcifying organisms?
- L297. It seems that Acropora drives the negative relation between wave energy and hard corals and calcifying organisms. If so, please clearly state that this is the case as no other groups or genus was negatively related with wave energy.

Figures
- Fig 5. Increase the font size of the graphs because they are hard to read even at 150% zoom, specially the genus name in each panel. Consider using abbreviations for genus names if PCoA plots are too busy.
- Fig 5. Why the center of the Spearman’s Rank correlation vectors are not departing from (0,0) for panels A and B as correctly show in panel C.
Supplementary Information
- Supplementary table 1. Row for Hard coral at the Kingman forereef should say 5m, se, 10m, se, 20m, se, instead of 10m, se, 10m, se, 10m, se.

Tables
- Table 1 and 2. Consider combining these two tables in one single table since the information is similar.
- Table 2. There are 11 genus instead 10 genus as stated in the table heading.
- Table 3. Pairwise comparisons appear for the first time in this table but it is not explained in the Methods. Please provide a brief account of how pairwise comparisons were done in the methods.

Discussion
- L357 What kind of debris? Coral skeletons from the crest and forereefs?
- L361 Substitute “predation” by “grazing”.
- L390-392 Reference(s) to support this statement.
- L420 Great point about discriminating between calcified and fleshy macroalgae when linking algae cover environmental factors.
- L448-451. As the author stated, the greatest wave energy is originated from the northwest at both atolls, thus Kingman’s triangle reef shape affords some protection, and sites at Kingman may experience a larger gradient of way energy than Palmyra. But, the west-east elongated shape of Palmyra might also provide differential protection from high wave energy on the south coast specially when there is also land blocking the waves. This is not explained in the manuscript and the reasons that Kingman “may experience a larger gradient of way energy” is unsupported by the data at hand.
- L460. Does sedimentation from the lagoon impact every forereef around Palmyra? How is the response of the benthic community to wave energy when those sediment-impacted sites are not considered in the analysis?

Experimental design

The methods, description of the study sites, and statistical analysis are appropriate to address the main objectives of this research. In the last paragraph of the methods, however, the authors should briefly explained why a dispersion-weighting procedure, which has been justified in the context of a generalized Poisson model for species counts, was adequate to analyze clustering of hard coral cover. Clark et al 2006 nicely explained the generality of this approach for hard bottom communities using photographic grids. Although the authors certainly cited this work a brief statement on the manuscript is necessary.

Validity of the findings

The data presented although descriptive is robust and statically sound. The discussion, however, is missing a final paragraph concluding or summarizing the main findings in the study and linking these findings to the bigger picture of how physical and oceanography factors structure remote and undisturbed benthic communities of reef atolls in the Pacific. The author’s research objectives are relevant to the field of coral reef ecology and conservation because few studies have analyzed how these natural factors are affecting undisturbed reef ecosystems in the absence of human activities. Thus, understanding these natural processes is important to discern how human activities have impacted the natural state of oceanic coral reef islands.

Additional comments

No Comments

Reviewer 3 ·

Basic reporting

This submission is logically organized and very clearly written, without extraneous material that does not add to the content or quality. Tables and Figures add to the understanding and visualization of the content narrated in the text. Figure 4, Stylized Reef Profiles, is particularly creative, with concisely illustrated and useful graphics.

Experimental design

The purpose of the paper, clearly expressed in the Introduction ("Our aim was to provide a comprehensive description of benthic community patterns across reef habitats, depths and, in particular, across horizontal wave energy gradients.") was properly addressed through in situ benthic surveys analyzed in conjunction with modeled wave data. The selected benthic survey methodology (photoquadrats) is one of several standards in the arena of coral reef ecology, and the acquisition of durable imagery from which quantitative data have been extracted would enable re-analysis, if desired. The sites are also amendable to resurvey in the future, hence the detailed data presented in Supplementary Table 1 provide a useful baseline for future comparisons.

Validity of the findings

The paper conforms to all bulleted criteria.

Additional comments

Adherence to all professional standards made this an enjoyable and interesting paper to review.

---

## Round 0.2 · accepted · Accept

Thank you for your fast and thorough response to the excellent reviewer comments and suggestions. I am really happy to see this paper in PeerJ.